# Metal Oxide Nanostructures Enhanced Microfluidic Platform for Efficient and Sensitive Immunofluorescence Detection of Dengue Virus

**DOI:** 10.3390/nano13212846

**Published:** 2023-10-27

**Authors:** Pareesa Pormrungruang, Supranee Phanthanawiboon, Sukittaya Jessadaluk, Preeda Larpthavee, Jiraphon Thaosing, Adirek Rangkasikorn, Navaphun Kayunkid, Uraiwan Waiwijit, Mati Horprathum, Annop Klamchuen, Tanapan Pruksamas, Chunya Puttikhunt, Takao Yasui, Mitra Djamal, Sakon Rahong, Jiti Nukeaw

**Affiliations:** 1College of Materials Innovation and Technology, King Mongkut’s Institute of Technology Ladkrabang, Chalongkrung Rd., Ladkrabang, Bangkok 10520, Thailand; 63607003@kmitl.ac.th (P.P.); 62607006@kmitl.ac.th (S.J.); 65116008@kmitl.ac.th (P.L.); adirek.ra@kmitl.ac.th (A.R.); navaphun.ka@kmitl.ac.th (N.K.); jiti.nu@kmitl.ac.th (J.N.); 2Department of Microbiology, Faculty of Medicine, Khon Kaen University, Khon Kaen 40002, Thailand; supraph@kku.ac.th (S.P.); jiraphon_thaosing@kkumail.com (J.T.); 3National Electronics and Computer Technology Center, National Science and Development Agency, Pathumtani 12120, Thailand; uraiwan.waiwijit@nectec.or.th (U.W.); mati.horprathum@nectec.or.th (M.H.); 4National Nanotechnology Center, National Science and Development Agency, Pathumtani 12120, Thailand; annop@nanotec.or.th; 5National Center for Genetic and Engineering and Biotechnology (BIOTEC), National Science and Development Agency, Pathumtani 12120, Thailand; tanapan.pro@biotec.or.th (T.P.); chunyapk@biotec.or.th (C.P.); 6Department of Life Science and Technology, Tokyo Institute of Technology, B2-521, 4259 Nagatsuta-cho, Midori-ku, Yokohama 226-8501, Japan; yasuit@bio.titech.ac.jp; 7Department of Physics, Faculty of Mathematics and Natural Sciences, Bandung Institute of Technology, Bandung 46132, Indonesia; mitra.djamal@yahoo.co.id

**Keywords:** ZnO nanorods, microfluidic platform, herringbone structure, Dengue virus, immunofluorescence

## Abstract

Rapid and sensitive detection of Dengue virus remains a critical challenge in global public health. This study presents the development and evaluation of a Zinc Oxide nanorod (ZnO NR)-surface-integrated microfluidic platform for the early detection of Dengue virus. Utilizing a seed-assisted hydrothermal synthesis method, high-purity ZnO NRs were synthesized, characterized by their hexagonal wurtzite structure and a high surface-to-volume ratio, offering abundant binding sites for bioconjugation. Further, a comparative analysis demonstrated that the ZnO NR substrate outperformed traditional bare glass substrates in functionalization efficiency with 4G2 monoclonal antibody (mAb). Subsequent optimization of the functionalization process identified 4% (3-Glycidyloxypropyl)trimethoxysilane (GPTMS) as the most effective surface modifier. The integration of this substrate within a herringbone-structured microfluidic platform resulted in a robust device for immunofluorescence detection of DENV-3. The limit of detection (LOD) for DENV-3 was observed to be as low as 3.1 × 10^−4^ ng/mL, highlighting the remarkable sensitivity of the ZnO NR-integrated microfluidic device. This study emphasizes the potential of ZnO NRs and the developed microfluidic platform for the early detection of DENV-3, with possible expansion to other biological targets, hence paving the way for enhanced public health responses and improved disease management strategies.

## 1. Introduction

Emerging infectious diseases, often caused by viruses, continue to pose significant public health threats worldwide. In the past few decades, we have seen the emergence and re-emergence of several viral pathogens that have led to significant global health crises. Notable examples include the Ebola virus, SARS, MERS, Zika, and most recently, SARS-CoV-2, the causative agent of COVID-19 [1,2,3,4]. The catastrophic impact of these outbreaks has highlighted the urgent need for effective early detection systems, which are vital for the timely initiation of treatment and to curb the spread of infection [5,6,7]. Early detection and diagnosis of viral infections remain challenging due to the ability of pathogens to evolve and adapt, resulting in more virulent and drug-resistant strains [8,9,10]. Conventional diagnostic methods such as viral culture, PCR, and serology often require skilled personnel, sophisticated laboratory equipment, and are time-consuming, making them unsuitable for immediate, point-of-care diagnosis [11,12]. 

In previous research, nanomaterials have been employed for Dengue fever detection using both Electrochemical [13] and Surface Plasmon Resonance (SPR) [14] methods. The enhanced sensitivity of these methods is attributed to the high surface area of nanomaterials, with specific studies even utilizing graphene structures [15] to augment the surface area for biomolecule interactions. However, these techniques still bear certain drawbacks, including depletion of reagents, time-consuming procedures requiring extensive instrumentation, specialist knowledge for operation, and lack of portability. Hence, integrating nanomaterials with microfluidic systems for pathogen detection could circumvent these limitations and broaden applications in pathogen detection.

As a response to these challenges, microfluidic platforms have emerged as promising tools for rapid, accurate, and portable viral detection [16,17,18,19]. Microfluidic platforms, by their design, allow for the miniaturization of complex laboratory processes, significantly reducing sample and reagent consumption, and shortening the total assay time [20]. These features are particularly advantageous in resource-constrained settings, making microfluidic platforms a potent tool for widespread, point-of-care diagnostics. Incorporating nanostructures into these devices can significantly augment their biosensing capabilities, leading to enhanced sensitivity and specificity [21,22,23]. Zinc Oxide nanorods (ZnO NRs), in particular, have gained substantial attention in the field of biosensing. These nanostructures exhibit a high surface-to-volume ratio, excellent biocompatibility, and unique optical and electronic properties [15,18,21,24,25,26]. Studies have shown that ZnO NRs can provide abundant binding sites for biomolecules, thereby enhancing the immobilization capacity and the detection sensitivity [24,26,27,28]. Recent advancements have also demonstrated the potential of ZnO NRs in the field of optoelectronics, photocatalysis, and as gene delivery vectors, opening up a range of potential applications [29,30].

The integration of ZnO NRs into microfluidic devices has been shown to augment biosensing performance [26,28,31,32,33,34]. The combined benefits of rapid, small-scale fluid handling and high-performance nanomaterials have the potential to revolutionize point-of-care diagnostics. Moreover, by using a surface functionalization strategy such as (3-Glycidyloxypropyl)trimethoxysilane (GPTMS) modification, the biosensing capability of these platforms can be further enhanced. Studies have reported that such surface modifications can improve the stability and selectivity of the biosensor by providing covalent binding sites for specific biomolecules [26].

The current study aims to synthesize ZnO NRs using a seed-assisted hydrothermal synthesis method and assess the functionalization efficiency on ZnO NRs and bare glass substrates. We investigate the crystal structure of ZnO NRs using X-ray diffraction (XRD) patterns. These patterns provide insights into the arrangement of atoms within the ZnO NRs. Field emission–scanning electron microscopy (FE-SEM) is a technique used to obtain the morphology images of surfaces. In addition, FT-IR was employed to study the interaction between Zinc Oxide nanorods (ZnO NRs), GPTMS, and antibodies. The spectra were obtained in transmission mode, which allows the investigation of chemical bonds and functional groups present in the sample by analyzing how it absorbs infrared light. We also focus on the development and optimization of an immunofluorescence assay integrated with a microfluidic platform for efficient DENV-3 detection. Our findings contribute to the understanding of ZnO NR-based biosensing platforms and offer a robust, sensitive, and cost-effective strategy for rapid detection of DENV-3. Additionally, the strategies outlined in this study could potentially be extended to other biosensing applications. The ultimate goal is to create a device that offers high sensitivity and specificity in a miniaturized, portable, and easy-to-use format, thereby fulfilling the needs of point-of-care diagnostics. These advancements could significantly enhance our capacity to respond to emerging infectious diseases, providing an invaluable tool for global health.

## 2. Materials and Methods

### 2.1. Dengue Virus and Monoclonal Antibodies Preparation

The Dengue virus serotype 3 (DENV-3) was clinically isolated from a Thai patient in 2008. The virus was propagated in Vero cells before being subjected to titration by a focus-forming assay [35]. Briefly, Vero cells (derived from the kidney of the African green monkey) were cultured at 37 °C in Dulbecco’s Modified Eagle Medium (DMEM) (Gibco, Thermo Fisher Scientific, Inc., Waltham, MA, USA), supplemented with 10% fetal bovine serum (FBS) (HyClone Laboratories Inc., Logan, UT, USA). These cells were prepared as a monolayer in a 96-well plate before being infected with the serially diluted virus. The virus was then titrated by a focus-forming assay (FFA), a traditional method for quantifying infective viral particles. All methods performed in this study used the same lot of the virus, prepared from Vero cells, with an infectious particle concentration of 7.86 × 10^5^ FFU/mL and a total RNA concentration of 31.1 ng/µL.

For antibody preparation, two clones of monoclonal antibodies, originating from humans and mice, and specific to the envelope protein of Dengue, were prepared for this study. Briefly, hybridoma cells were thawed from −80 °C and cultured in RPMI medium (Thermo Fisher Scientific, Inc. Waltham, MA, USA) supplemented with 10% FBS at 37 °C, with 5% CO_2_. After four days of incubation, the antibody from the cell culture supernatant was collected and pooled in a sterile bottle. The pooled culture supernatant was stored at 4 °C until purification. Antibody purification was performed using a HiTrap Protein A HP column (Code No.17-0403-03, Global Life Sciences, Solutions USA LLC, Marlborough, MA, USA) following the company instructions.

### 2.2. Captured ELISA

In-house captured ELISA was set to characterize and confirm the binding ability of the antibodies used in this study [36,37]. The Dengue envelope-specific human monoclonal antibody 1 µg/mL was coated overnight at 4 °C with the coating buffer. DENV-3 at various dilutions from undiluted to 10^−7^ was incubated with the coated plate at room temperature for 1 h followed by dengue envelope-specific mouse monoclonal antibody 1 µg/mL and HRP labeled anti-mouse IgG (Abcam plc, Cambridge, UK). Color visualization was developed with TMB (Biolegend company, CA, USA) and the adsorbent results were measured by a microplate reader (Varioskan LUX, Thermo Fisher Scientific, Waltham, MA, USA). The cutoff for positive value will be calculated by mean negative plus 3SD [38]. The result showed that the captured ELISA method developed here can detect the virus with 78.6 FFU/mL (or 31.1 × 10^−4^ ng/µL) (Appendix A).

### 2.3. Synthesis of ZnO Nanorods via Hydrothermal Growth Method

Zinc oxide nanorods (ZnO NRs) were synthesized on glass substrates via the hydrothermal growth method. Initially, a ZnO seed solution was prepared by dissolving 25 mM of zinc acetate (Sigma-Aldrich, Inc., MO, USA) in 50 mL of isopropanol. This solution was continuously stirred at 60 °C for 1 h. Subsequently, diethanolamine (Sigma-Aldrich, Inc., MO, USA) was added to the solution, which was then stirred for an additional 2 h. The seed layer was deposited onto cleaned glass substrates (Paul Marienfeld GmbH, Lauda-Königshofen, Germany) via spin-coating (SPIN 150, SPS international, Putten, The Netherlands) the seed solution at 1000 rpm for 30 s. This layer was annealed at 350 °C for 4 h to improve adhesion and crystallinity. Following the seed layer preparation, the hydrothermal growth method was used to synthesize ZnO NRs. This method was adapted and modified from the previously published literature [29,39]. The growth solution was composed of 25 mM of zinc nitrate hexahydrate (Zn(NO_3_)_2_·6H_2_O) and hexamethylenetetramine (HMTA) (Sigma-Aldrich, Inc., MO, USA) in deionized water. The glass substrate with the ZnO seed layer was immersed in the growth solution, and the reaction was conducted at 90 °C for 6 h. The substrate was then rinsed thoroughly with deionized water and dried using a gentle stream of nitrogen gas.

### 2.4. Conjugation of Monoclonal Antibodies (mAbs) onto ZnO Nanorods 

Figure 1A is the schematic of the sZnO NRs surface functionalization for the immunofluorescence assay. A silanization agent, (3-glycidyloxypropyl)trimethoxysilane (GPTMS) (Sigma-Aldrich, Inc., MO, USA), was used. To prepare the antibody conjugation, ZnO NRs substrates were immersed into 4% (*v*/*v*) solution of GPTMS in ethanol for 2 h. Subsequently, ZnO NRs substrates were rinsed with ethanol and ultrapure water to remove any unreacted GPTMS. The capture antibody was introduced into the ZnO NRs surface and placed in a controlled-humidity environment within a glass container, ensuring optimal conditions for antibody conjugation. The mouse anti-flavivirus envelope protein antibody (4G2 mAb) was allowed to coat the ZnO NRs’ surface for 30 min through epoxy ring-opening reactions between the antibody and the GPTMS. The functionalized ZnO NR substrates were then washed with phosphate-buffered saline (PBS) to remove any unbound antibodies. To minimize nonspecific adsorption, the surface was blocked with a 5% (*w*/*w*) bovine serum albumin (BSA) solution (Sigma-Aldrich, Inc., MO, USA) for 30 min. The ZnO NR substrates were subsequently washed with a solution containing 0.1% Tween-20 (Sigma-Aldrich, Inc., MO, USA) in PBS to further reduce nonspecific binding. Lastly, 0.5 µg/mL of a goat anti-mouse IgG tagged with Alexa Fluor^®^ 488) (Abcam, Inc., Cambridge, UK) in PBS buffer was introduced to evaluate the conjugation of the antibody on the ZnO NRs’ surface by the visualization of the captured antibodies through an indirect immunoassay.

### 2.5. Microfluidic Chip Design and Fabrication

Microfluidic chips were designed employing KLayout Layout Viewer and Editor (version 0.27.7, Matthias Köfferlein), and fabricated using a soft lithography technique based on previously reported procedures [40,41,42]. The photolithography process was initiated by preparing the photomasks, consisting of a 500 µm width microchannel pattern and a herringbone pattern [43]. A layer of SU-8 3050 negative photoresist (Kayaku Advanced Materials, Inc., Westborough, MA, USA), with a thickness of 60 µm, was spin-coated onto a silicon wafer at a speed of 500 rpm for 15 s, and then at 1400 rpm for 30 s. The coated wafer underwent a soft bake at 65 °C for 1 min, followed by a post-exposure bake at 95 °C for 5 min. The microchannel pattern was transferred onto the photoresist by exposing it to UV light using a mask aligner (MA-10, Mikasa Co., Ltd., Tokyo, Japan). The unexposed regions were removed by immersing the wafer in a developer solution for 5 min, and subsequently rinsing with isopropanol. For fabricating the herringbone structure, an additional layer of SU-8 3050 was spin-coated onto the microchannel mold. Similar to the earlier step, the wafer was subjected to baking and exposure to UV light. The unexposed regions were removed using a developer solution, followed by rinsing with isopropanol. The microchannel mold was further protected via silanization using Trichloro(1H,1H,2H,2H-perfluorooctyl)silane (Sigma-Aldrich Inc., MO, USA) under vacuum overnight. The final mold was replicated by casting polydimethylsiloxane (PDMS) (SYLGARD 184 Silicone Elastomer Kit, Dow Inc., Midland, MI, USA), prepared by mixing elastomer and curing agent in a 10:1 weight ratio, over the SU-8 mold. This was cured at 80 °C for 2 h. The PDMS replica was carefully peeled from the SU-8 mold and then bonded onto the glass substrate containing the synthesized ZnO NRs through a standard PDMS bonding technique [44].

### 2.6. Sensitivity of the DENV-3 Quantitative Assay Using Immunofluorescence on ZnO-Nanorod-Integrated Microfluidic Platform

The quantitative assay for Dengue virus serotype 3 (DENV-3) detection was conducted based on an indirect immunofluorescence assay, utilizing a microfluidic platform integrated with ZnO NRs, as shown in Figure 1B. Initially, the ZnO NR surfaces were functionalized and modified by introducing a 4% (*v*/*v*) GPTMS solution in ethanol at a flow rate of 120 µL/min for 2 h. Subsequently, the channels were rinsed with ethanol and ultrapure water to remove excess GPTMS. Conjugation with 1 µg/mL solution of 54 monoclonal antibody (54 mAb) was performed by introducing at a flow rate of 120 µL/min for 2 min. Subsequently, flow was halted, and the sample was incubated at room temperature for 30 min. The system was then washed with 0.1% Tween-20 in phosphate-buffered saline (PBST) (Sigma-Aldrich, Inc., MO, USA) at a flow rate of 120 µL/min for 2 min. This was followed by a nonspecific binding step using a 5% (*w*/*w*) bovine serum albumin (BSA) solution, also at a flow rate of 120 µL/min for 2 min. Another incubation period of 30 min was implemented before washing the system again with 0.1% PBST for 2 min. The Dengue virus serotype 3 (DENV-3) was then introduced at varying concentrations, ranging from 3.1 × 10^−4^ to 3.1 × 10^1^ ng/mL, with a consistent flow rate of 120 µL/min for 2 min. This was followed by a 30 min incubation period and a subsequent wash with 0.1% PBST for 2 min. The 4G2 monoclonal antibody was then flowed at a concentration of 1 µg/mL and a flow rate of 120 µL/min for 2 min, followed by a 30 min incubation. The system was then washed with 0.1% PBST for 2 min. Finally, the goat anti-mouse IgG tagged with Alexa Fluor^®^ 488 (Abcam, Inc., Cambridge, UK) was introduced at a flow rate of 120 µL/min for 2 min, followed by a 15 min incubation period and a final wash with 0.1% PBST for 2 min to remove any unbound secondary antibodies. The chip was then ready for fluorescence imaging. Fluorescence images were captured using an inverted fluorescence microscope (Oxion Inverso, Euromex Microscopen BV, Arnhem, The Netherlands). These images were processed and analyzed using ImageJ software (1.53K Java 1.8.0_172, https://imagej.nih.gov) to quantify the presence of DENV-3.

### 2.7. XRD Analysis

X-ray diffraction (XRD) (Smart lab, Rigaku Corp., Tokyo, Japan) using Cu-Kα radiation (λ = 1.54 Å) is a technique used to investigate the crystal structure of ZnO seed layer and ZnO NRs on glass substrates. A diffractometer was employed to acquire diffraction patterns at angles ranging from 20° to 60° and analyzed by Rigaku Data Analysis Software, PDXL 2, Versions 2.4.2.0.

### 2.8. SEM Imaging

FE-SEM (Apreo 2S, Thermo Fisher Scientific, Inc., MA, USA), which operated at the acceleration voltage of 10 kV, was utilized to capture images of synthesized Zinc Oxide nanorods (ZnO NRs) on glass slides. The dimensions of ZnO NRs were measured by ImageJ software (https://imagej.nih.gov).

### 2.9. FT-IR Analysis

FT-IR measurements (Spectrum Two FT-IR Spectrometer, PerkinElmer Inc., CT, USA) were carried out in the wave number range of 400–4000 cm^−1^. All spectra were measured with a minimum of 8 scans. These data are essential for understanding the nature of the interactions between ZnO, GPTMS, and antibodies.

## 3. Results and Discussion

### 3.1. The Seed-Assisted Hydrothermal Synthesis Method for ZnO Nanorods and Their Characterization

A seed-assisted hydrothermal synthesis method was employed to synthesize ZnO NRs on meticulously cleaned glass slides. The crystalline structure of both the seed layer and the synthesized ZnO NRs was investigated using X-ray diffraction (XRD), as depicted in Figure 2A. The XRD spectra of the ZnO NRs on the glass substrate exhibit characteristic peaks, with a particularly prominent peak corresponding to the (002) plane. This is consistent with the standard XRD pattern of hexagonal wurtzite ZnO (JCPDS card No. 36-1451). Notably, the absence of any impurity peaks in the XRD pattern suggests the high purity and crystalline of the synthesized ZnO NRs. The ZnO NRs exhibit a preferred growth direction along the c-axis, which is a consequence of the seed-assisted hydrothermal synthesis method. This preferred orientation is illustrated in the SEM images shown in Figure 2B,C. Morphologically, the ZnO NRs are characterized by a high surface-to-volume ratio, which is advantageous for the purpose of biosensing, as it offers an abundance of binding sites for the conjugation of capture antibodies, DNA aptamers, peptides, and other biomolecules. The distribution and diameter of the ZnO NRs were measured and analyzed using ImageJ software (https://imagej.nih.gov). The histogram depicting the size distribution of the ZnO NRs is presented in Figure 2D.

### 3.2. Comparative Analysis of Functionalization Efficiency on ZnO Nanorod Substrate and Bare Glass Substrate

We conducted an assessment of the efficiency of conjugation of 4G2 monoclonal antibody (mAb) on ZnO NR surfaces in comparison to bare glass substrates. During the functionalization process, GPTMS was used in ethanol at concentrations of 4% and 8% as a covalent linker. This facilitated the binding between the 4G2 mAb and the surfaces under investigation. Fourier-transform infrared spectroscopy (FT-IR) was employed to examine the chemical functionalities introduced onto the GPTMS-modified ZnO NRs on the glass substrate. The FT-IR spectrum, as illustrated in Figure 3, reveals several distinct absorption bands. There is a broad absorption band observed around 2974 and 2940 cm⁻¹, which is generally attributed to the stretching vibrations of CH_3_ groups. An additional absorption feature is visible at roughly 2840 cm⁻¹, attributed to the first overtone of the hydroxyl group stretching mode. Furthermore, absorption bands are present at 1080 and 878 cm⁻¹, which are typically ascribed to Si-CH_2_ and Si-O vibrations, respectively [26,45,46,47]. The hydroxyl group presence is especially noteworthy as it constitutes a reactive site that can be modified with a variety of silane reagents carrying different functional groups, such as amino (-NH_2_), thiol (-SH), and alkyne (-C≡C-). This chemical versatility is beneficial for conjugating biomolecules like antibodies to the surface. The FT-IR spectra of 4G2 mAb bound to GPTMS-modified ZnO NRs are depicted in Figure 3A. We observed absorption bands for CN stretching combined with NH bending and C=O stretching at 1542 and 1653 cm⁻¹, respectively.

Subsequently, we deployed the goat anti-mouse IgG tagged with Alexa Fluor^®^ 488 to illuminate the interaction dynamics between the 4G2 mAb and the ZnO NRs, which were modified with 3-glycidoxypropyltrimethoxysilane (GPTMS). The fluorescence microscopy was conducted under rigorously controlled conditions to assure uniformity. The captured fluorescence images were subsequently analyzed to quantitatively evaluate the effectiveness of mAb surface modification. As illustrated in Figure 4, the comparative analysis of fluorescence intensities from the bare glass slides, non-modified ZnO NRs, GPTMS-modified glass slides, 4% GPTMS-modified ZnO NRs, and 8% GPTMS-modified ZnO NRs delivered insightful results. The 4% GPTMS-modified ZnO NRs exhibited fluorescence intensities that were 2.68-fold, 2.21-fold, and 1.27-fold higher than those of bare glass slides, ZnO nanostructured glass slides, and GPTMS-modified glass slides, respectively. These findings emphasize the signal-amplifying capacity of ZnO NRs, even in the absence of GPTMS modification. In particular, the ZnO substrates exhibited significant fluorescence after a 30 min incubation period with the mAb. Interestingly, the 8% GPTMS-modified ZnO NRs exhibited fluorescence intensities that were comparable to those of the ZnO NRs substrate. This could potentially be attributed to the higher concentration of the covalent linker resulting in an over-modification of the surface, leading to a steric hindrance effect that could negatively impact the antibody binding process [48,49]. In these findings, and after considering the consistent of the 4% GPTMS-modified ZnO NRs, we selected them as the most suitable substrate for the subsequent viral detection experiments.

### 3.3. Optimization and Validation of a ZnO-NR-Surface-Integrated Microfluidic Platform for the Immunofluorescence Detection of DENV-3

In light of the maximal fluorescence intensity observed with the 4% GPTMS-modified ZnO NRs substrate, this configuration was selected for the fabrication of an integrated microfluidic platform. Incorporation of the optimally conditioned substrate within the microchannel, featuring a herringbone structure, was accomplished via a PDMS bonding method, as presented in Figure 5A,B. The immunofluorescence assay was designed to function on this optimized ZnO NRs microfluidic platform, as outlined in Figure 1B. Post modification of the surface with 4% GPTMS and clone 54 mAb, non-specific binding was mitigated using a BSA blocking solution. Serial dilutions of DENV-3, ranging from 3.1 × 10¹ ng/mL to 3.1 × 10^−4^ ng/mL, were then introduced to the platform. This led to the successful capture of antibody–antigen pairs prior to the loading of the 4G2 mAb and fluorescent dye onto the ZnO NRs microfluidic platform. The viruses were detected via the immunofluorescence assay on the GPTMS-modified ZnO NRs surface within the microchannel. The fluorescence signal was generated using the goat anti-mouse IgG tagged with Alexa Fluor^®^ 488 leveraging its high affinity. The ZnO nanostructured surface, characterized by its extensive roughness, provides an abundant number of binding sites for DENV-3, while simultaneously reducing the diffusion distance required for the target and reagent solutions. 

The fluorescence images corresponding to various virus concentrations, including a negative control, are illustrated in Figure 5A. Analysis of the images reveals a discernible decrease in intensities with declining DENV-3 concentration. This trend is substantiated by the quantitative data collected from the immunofluorescence assay, presented in Figure 5B, wherein the dynamic detection range is noted to span from 3.1 × 10^3^ to 3.1 × 10^−4^ ng/mL. The herringbone structure inherent in the microfluidic platform facilitates enhanced fluidic mixing, thereby augmenting the interaction probability between the DENV-3 and the immobilized 4G2 mAb on the ZnO surface. This feature allows for rapid detection in a minimal sample volume, thereby increasing the specificity of our assay. As conveyed by the dose–response curve in Figure 5B, an increase in the viral concentration directly corresponds to an increase in fluorescence intensity. The limit of detection (LOD) for the DENV-3 immunofluorescence assay, implemented on the ZnO NRs within the microfluidic platform, was ascertained to be 3.1 × 10^−4^ ng/mL, which is five-fold higher than the intensity recorded for the negative control. The measured fluorescence exhibited clear distinction from background fluorescence, suggesting a significant signal-to-noise ratio. A comparative analytical performance of our ZnO NRs integrated microfluidic platform with the previous reports on the topic of Dengue virus detection and diagnostic is presented in Table 1.

The pronounced sensitivity of this ZnO-nanostructure-based immunofluorescence microfluidic platform can be ascribed to several factors. Firstly, the ZnO-NR-modified surface affords an increased density of binding sites for the mAbs in comparison to a flat glass surface. Secondly, the diffusion distance between the surface and the species in the flow (such as viral samples, mAb, and reagents) is minimized. Lastly, the enhancement in fluorescence intensity can be attributed, in part, to the increased antigen–antibody pairing per unit substrate area facilitated by the larger surface area of the ZnO NRs. We propose that the implementation of an immunofluorescence assay on a ZnO-NR-surface-integrated microfluidic platform holds promise for diverse biosensing applications, such as the detection of nucleic acids, proteins, and other biologically relevant targets. Furthermore, the adaptability of this immunofluorescence platform invites its potential integration with alternative detection methods.

## 4. Conclusions

In conclusion, our research has provided a detailed exploration into the synthesis, functionalization, and application of zinc oxide nanorods (ZnO NRs) for the detection of DENV-3. The seed-assisted hydrothermal synthesis method has proved to be a viable approach to produce high-purity ZnO NRs, with a prominent growth orientation that benefits biosensing applications due to a high surface-to-volume ratio. These characteristics offer an abundance of binding sites for the conjugation of biomolecules, key to our detection mechanism. Functionalization with 4G2 monoclonal antibody (mAb) was significantly more efficient on ZnO NRs surfaces compared to bare glass substrates. This was demonstrated through comparative fluorescence intensities, with the 4% GPTMS-modified ZnO NRs exhibiting up to 2.68-fold greater intensities than bare glass slides. Importantly, this highlights the signal-amplifying capacity of ZnO NRs, crucial for sensitive detection. Our final fabrication of a ZnO-NR-integrated microfluidic platform capitalized on these findings, utilizing the 4% GPTMS-modified ZnO NRs for their optimal fluorescence intensity. The immunofluorescence detection of DENV-3 on this platform demonstrated a significant detection range, from 3.1 × 10^3^ to 3.1 × 10^−4^ ng/mL. This sensitivity marks an important achievement in the quest for early, reliable detection of DENV-3. Moreover, the potential versatility of this ZnO-NR-integrated platform is promising, with potential applications beyond DENV-3 detection. With further refinement and validation, this platform could play a significant role in a diverse array of biosensing applications, including the detection of various nucleic acids, proteins, and other biologically relevant targets. Therefore, it is our belief that the present study offers a substantial contribution to the advancement of biosensing technology, particularly in relation to the early detection of infectious diseases.

## Figures and Tables

**Figure 1 nanomaterials-13-02846-f001:**
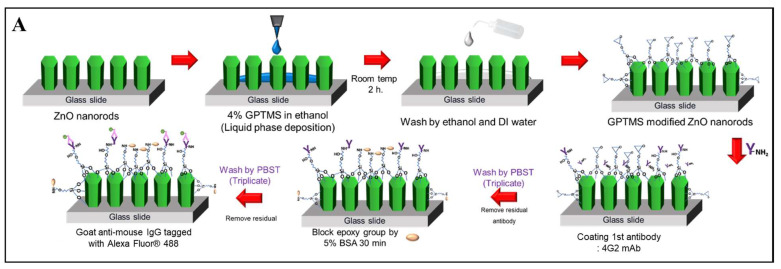
An illustrative depiction of the process involved in surface functionalization and the subsequent immunofluorescence assay of Dengue virus serotype 3 (DENV-3). (**A**) A schematic demonstrating the surface functionalization process using 4% GPTMS in ethanol on Zinc Oxide nanorods’ (ZnO NRs) surface. This is followed by conjugation with the monoclonal antibody 4G2 (4G2 mAb), setting the stage for the indirect immunoassay. (**B**) A complementary schematic outlining the procedure for the immunofluorescence assay of DENV-3, demonstrating the subsequent steps in the testing process.

**Figure 2 nanomaterials-13-02846-f002:**
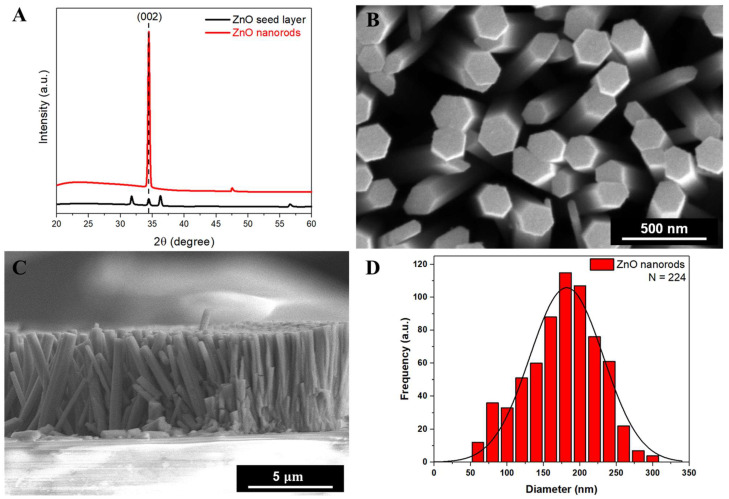
Comprehensive characterization of the Zinc Oxide nanorods (ZnO NRs) synthesized on glass slides. (**A**) An X-ray diffraction (XRD) pattern of the ZnO NRs grown on a glass substrate and ZnO seed layer. (**B**) An enlarged plan-view field emission–scanning electron microscopy (FE-SEM) image showcasing the morphology of the ZnO NRs. (**C**) A cross-sectional FE-SEM image that offers a detailed look at the density and orientation of the ZnO NRs, providing insights into their vertical growth pattern. (**D**) A histogram illustrating the distribution of diameters among the synthesized ZnO NRs, as analyzed and determined using ImageJ software (https://imagej.nih.gov).

**Figure 3 nanomaterials-13-02846-f003:**
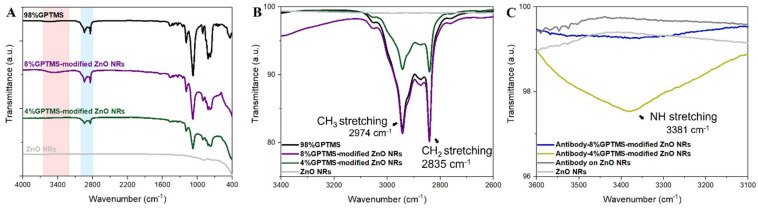
Fourier-transform infrared (FT-IR) spectroscopy analysis of ZnO nanorods (NRs) and their modifications. (**A**) The FT-IR spectra of unmodified ZnO NRs, ZnO NRs modified with 4% glycidoxypropyltrimethoxysilane (GPTMS), ZnO NRs modified with 8% GPTMS, and the 98% GPTMS solution, providing insights into their chemical functionalities. (**B**) The FT-IR spectra specifically showing the stretching vibrations of CH_3_ and CH_2_ groups in 4% GPTMS-modified ZnO NRs and 8% GPTMS-modified ZnO NRs. (**C**) The FT-IR spectra demonstrating the binding of 4G2 monoclonal antibody (mAb) with both 4% GPTMS-modified ZnO NRs and 8% GPTMS-modified ZnO NRs, indicating successful bioconjugation.

**Figure 4 nanomaterials-13-02846-f004:**
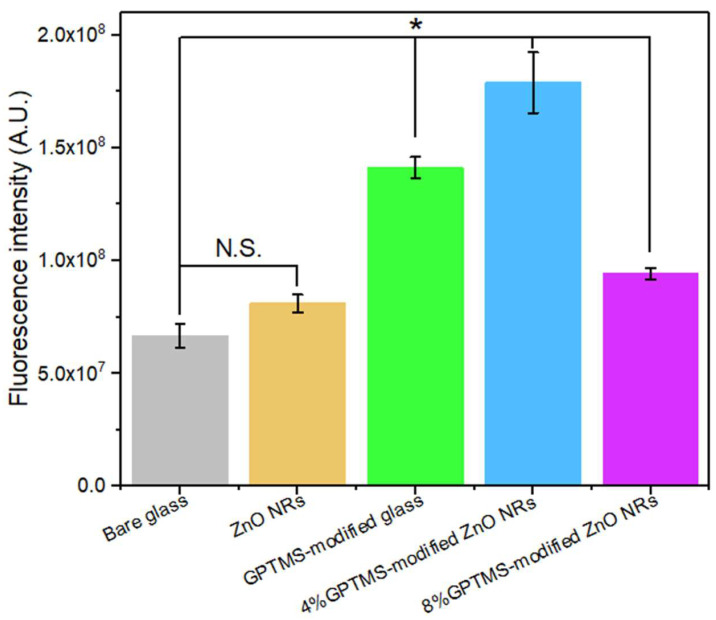
Comparative analysis of fluorescence intensities corresponding to the binding of Alexa Fluor 488 to the 4G2 monoclonal antibody (mAb) on various surfaces. The surfaces under comparison include bare glass, unmodified ZnO nanorods (NRs), 3-glycidyloxypropyl trimethoxysilane (GPTMS)-modified glass, ZnO NRs modified with 4% GPTMS, and ZnO NRs modified with 8% GPTMS. The observed fluorescence intensities reflect the efficiency of antibody conjugation under different GPTMS concentrations. The One-way ANOVA with post-HOC Tukey was used as the statistical test. N.S is non-significant and * *p* < 0.001. Error bars represent standard deviations derived from three independent samples for each substrate, indicating experimental reproducibility and the distribution of measured values.

**Figure 5 nanomaterials-13-02846-f005:**
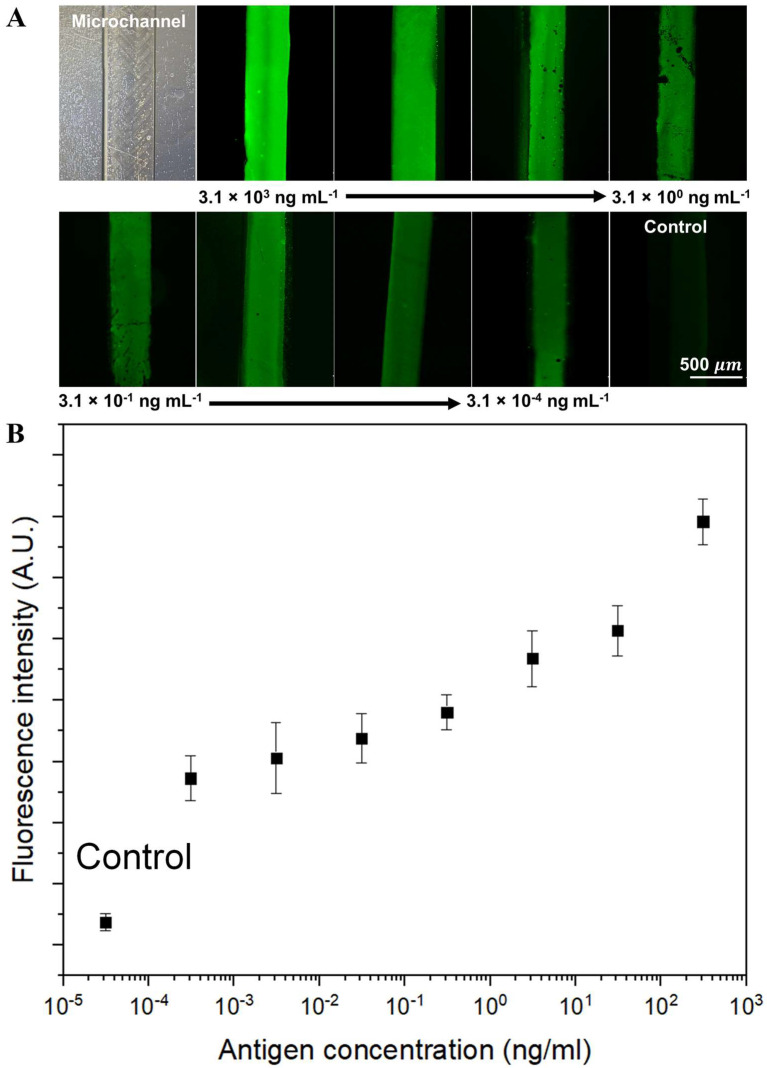
Detection of Dengue virus serotype 3 (DENV-3) via an on-chip indirect immunofluorescence assay using the ZnO-NR-integrated microfluidic platform. (**A**) A series of fluorescence microscopy images that demonstrate the on-chip immunofluorescence detection of DENV-3 at various concentrations, ranging from 3.1 × 10^3^ to 3.1 × 10^−4^ ng/mL by ten times dilution for each step. These images exemplify the robust detection capabilities of the microfluidic platform across a wide range of viral concentrations. (**B**) A graphical representation of the corresponding fluorescence intensities plotted against the varying concentrations of DENV-3 on a logarithmic scale. This plot highlights the proportional relationship between fluorescence intensity and viral concentration. Error bars represent the standard deviations calculated from three distinct experimental runs, affirming the consistency and reliability of the detection method.

**Table 1 nanomaterials-13-02846-t001:** Comparison of different methods for the detection of the Dengue virus.

Detection Method	Analyte	Matrix	Detection Method	Detection Range	Detection Limit	Detection Time	Ref.
Electrochemical impedance spectroscopy for dengue virus biosensor	Dengue serotype type 2	-	IgG antibody	1–900 PFU mL^−1^	^-^	-	[13]
Magnetic beads on a microfluidic system	Dengue serotype type 2	serum	Antibody	10^1^–10^6^ PFU mL^−1^	10^2^ PFU mL^−1^	-	[50]
A microfluidic dielectrophoresis platform	Dengue serotype type 2	-	Anti-DENV envelope protein antibody	0–10^6^ PFU mL^−1^	10^4^ PFU mL^−1^	5 min (pre-fabricated immunoassay is available)	[20]
SPR biosensor	Dengue serotype 2 and 3	serum	Monoclonal anti-flavivirus antibodies	-	2 × 10^4^ particles. mL^−1^	30 min (pre-fabricated immunoassay is available)	[14]
Biosensor based on SiNW/AuNP-modified screen-printed electrode	Dengue serotype type 1,2,3 and 4	-	DNA	1.0 × 10^−11^–1.0 × 10^−7^ M	1.63 × 10^−12^ M	10 min 35 s	[51]
SPCE-portable NS1-based electrochemical immunosensor	Dengue virus NS1	serum	Anti-NS1 monoclonal antibody	1–200 ng mL^−1^	0.30 ng mL^−1^	Assay time is not specified	[52]
SERS-based lateral flow biosensor	Dengue nonstructural protein 1	-	Polyclonal primary anti NS1antibody	15–500 ng mL^−1^	15 ng mL^−1^	Assay time is not specified	[23]
Graphene-SPCE	Dengue virus antibodies	serum	Envelope glycoprotein domain III (EDIII) antigen	125–2000 ng mL^−1^	22.50 ng mL^−1^	Assay time is not specified	[15]
Microfluidic paper-based analytical devices (μPADs)	Dengue NS1	serum	Anti-NS1 monoclonal antibody	0–1 μg mL^−1^	74.8 ng mL^−1^	20–30 min	[53]
Nanomaterial-enhanced microfluidic platforms	Dengue serotype type 3	Culture supernatants	Anti-DENV envelope protein monoclonal antibody	3.1 × 10^−4^–3.1 × 10^3^ ng mL^−1^	3.1 × 10^−4^ ng mL^−1^	15 min	This work

## Data Availability

Not applicable.

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
