# Peer review of "Metal Oxide Nanostructures Enhanced Microfluidic Platform for Efficient and Sensitive Immunofluorescence Detection of Dengue Virus"

_nanomaterials, 2023, doi:10.3390/nano13212846_

Round 1
Reviewer 1 Report
Authors synthesized ZnO NRs using a seed-assisted hydrothermal synthesis method and assess the functionalization efficiency on ZnO NRs and bare glass substrates. Thay also focus on the development and optimization of an immunofluores- 86 cence assay integrated with a microfluidic platform for efficient DENV-3 detection. Their findings contribute to the understanding of ZnO NRs based biosensing platforms and offer a robust, sensitive, and cost-effective strategy for rapid detection of DENV-3. Additionally, the strategies outlined in this study could potentially be extended to other biosensing applications. The ultimate goal is to create a device that offers high sensitivity and specificity in a miniaturized, portable, and easy-to-use format, thereby fulfilling the needs of point-of-care diagnostics. These advancements could significantly enhance our capacity to respond to emerging infectious diseases, providing an invaluable tool in the global health. It can be accepted as is.
Author Response
Dear Reviewer,
We are sincerely thankful for your insightful comments and the positive evaluation of our manuscript. Your summary encapsulates the core of our work and its potential impact on the development of biosensing platforms, particularly in the context of point-of-care diagnostics for emerging infectious diseases. We are optimistic that our study will pave the way for further research and development in this field, contributing to the global health response through innovative diagnostic tools.
We are committed to ensuring that our research is presented most accurately and comprehensively. We are grateful for your supportive comments and the time invested in reviewing our work.
Best regards,
Sakon

Reviewer 2 Report
Please, check the attached file

Author Response
Dear Reviewer,
We would like to express our sincere gratitude for the opportunity to revise our manuscript entitled " Metal Oxide Nanostructures Enhanced Microfluidic Platform for “Efficient and Sensitive Immunofluorescence Detection of Dengue Virus” Manuscript ID: nanomaterials-2646360. We are thankful to the reviewers for their constructive feedback, which has been instrumental in enhancing the quality and clarity of our work.
Below, we provide a point-by-point response to the referees' comments and detail the revisions made to the manuscript:
Reviewer-2
- Not a major issue, just a suggestion. I believe the arrow on the outlet port of Figure 1B should point in the opposite direction. Currently, the Inlet and Outlet arrows are pointing in the same direction.
Reply: We have improved Figure 1B to reflect the accurate flow direction through the microfluidic platform. The arrows correctly indicate the path from the inlet to the outlet, ensuring clarity in our system depiction. The revised figure has been incorporated into the manuscript.
- Some descriptions in the experimental sections might not be sufficiently clear for the reader. For instance, lines 169 to 172 in Section 2.4 seem to suggest that Alexa was introduced into the system, whereas Figure 1a depicts an anti-detector antibody tagged with Alexa. The text should be modified to indicate that what is included is an Alexa-tagged antibody.
Reply: We have revised the text in lines 169 to 172 to explicitly state that the system includes a goat anti-mouse IgG tagged with Alexa Fluor® 488. This clarification should align the text with Figure 1a and eliminate any ambiguity regarding the use of Alexa-tagged antibodies in our experimental setup.
- Section 2.6 needs improvement as some experimental details are either missing or unclear in the description. For instance, the sentences in lines 203 and 204, “Conjugation with a 1 μg/mL solution of 54 mAb was performed by introducing it at a flow rate of 120 μL/min. This was followed by two incubation steps at room temperature, each lasting 30 minutes,” lack clarity. How long was the 54 mAb flowing? Was the flow stopped during the incubation? How was the system washed between the two incubation steps? Similarly, the introduction of the DENV-3 sample is also unclear. How long was it flowing? Or, equivalently, what volume of the sample was injected? Does the incubation occur while the sample is flowing? Does the dynamic virus capture entail a continuous sample flow of 120 μL/min for 30 minutes, totaling 3.6 mL? Isn't that volume of the sample problematic?
Reply: We have revised this section to provide a more comprehensive and clear description of the experimental procedures (line 209-226). Specifically, we have clarified the duration of 54 mAb flow, whether the flow was stopped during incubation, and how the system was washed between the two incubation steps. We have also provided details on the introduction of the DENV-3 sample, including the flow duration, volume injected, and whether incubation occurs while the sample is flowing.
“Conjugation with 1 µg/mL solution of 54 monoclonal antibody (54 mAb) was performed by introducing at a flow rate of 120 µL/min for 2 minutes. Subsequently, flow was halted, and the sample was incubated at room temperature for 30 minutes. The system was then washed with a 0.1% Tween-20 in phosphate-buffered saline (PBST) (Sigma-Aldrich, Inc.) at a flow rate of 120 µL/min for 2 minutes. This was followed by a nonspecific binding step using a 5% (w/w) bovine serum albumin (BSA) solution, also at a flow rate of 120 µl/min for 2 minutes. Another incubation period of 30 minutes was implemented before washing the system again with 0.1% PBST for 2 minutes. The Dengue Virus serotype 3 (DENV-3) was then introduced at varying concentrations, ranging from 3.1×10-4 to 3.1×101 ng/ml, with a consistent flow rate of 120 µl/min for 2 minutes. This was followed by a 30-minute incubation period and a subsequent wash with 0.1% PBST for 2 minutes. The 4G2 monoclonal antibody was then flowed at a concentration of 1 µg/ml and a flow rate of 120 µl/min for 2 minutes, followed by a 30-minute incubation. The system was then washed with 0.1% PBST for 2 minutes. Finally, the goat anti-mouse IgG tagged with Alexa Fluor® 488 (Prima Scientific Co., Ltd.) was introduced at a flow rate of 120 µl/min for 2 minutes, followed by a 15-minute incubation period and a final wash with 0.1% PBST for 2 minutes to remove any unbound secondary antibodies. The chip was then ready for fluorescence imaging.”
- Sections 2.7, 2.8, and 2.9 appear to primarily describe the applications of these techniques, which may not be very pertinent in the Materials and Methods section. If the authors deem this information important, it might be better suited for the Introduction. Instead, Sections 2.7, 2.8, and 2.9 should focus on detailing the experimental and technical aspects of the measurements. For example, XRD descriptions typically include not only the type of radiation used but also the manufacturer and model of the diffractometer and the software employed for data analysis. Similarly, FTIR descriptions should specify the manufacturer and model of the equipment, the resolution, the number of scans, etc.
Reply: We have revised Sections 2.7, 2.8, and 2.9 to focus on the experimental and technical aspects of the measurements, as your suggestion. Information about the applications of these techniques has been moved to the Introduction section (line 86-92) for better context. Additionally, we have included specific details such as the type of radiation used, the manufacturer and model of the diffractometer, and the software employed for XRD data analysis. For FTIR, we have specified the manufacturer and model of the equipment, the resolution, and the number of scans.
- Figure 2b and 2c appear to be inconsistent in terms of size. According to the image scale of 500 nm in Figure 2b, the hexagons are about 250 nm from one vertex to the opposite one, which aligns with the histogram in Figure 2d. However, based on the scale in Figure 2c, each nanorod appears to be about 100 nm wide, which does not correspond with the values in Figure 2b nor with the histogram in Figure 2d. Could there be an explanation for this discrepancy?
Reply: Upon reviewing your comment, we found that the scale bar in Figure 2c was incorrect. We have revised the scalebar in Figure 2c to accurately reflect the dimensions of the nanorods, ensuring consistency with Figure 2b and the histogram in Figure 2d.
- From an academic standpoint, it would be advisable to avoid using the term 'nanorods' in the manuscript and replace it with 'microrods'. The average diameter is around 200 nm, and the height of these 'nanorods', according to Figure 2c, is about 500 nm (half a micron!). These structures, seemingly measuring 0.2 μm x 0.5 μm, are decidedly too large to be classified as nanomaterials. According to a common definition, a material should have at least one dimension below 100 nm to be considered a nanomaterial [https://euon.echa.europa.eu/definition-of-nanomaterial].
Reply: We appreciate your attention to the terminology used to describe the rod structures in our study. Our structures have a diameter that ranges from 50-300 nm and a length of 5 microns, resulting in an aspect ratio of about 1:10. According to the definitions and guidelines we have followed, these dimensions and aspect ratios are consistent with what is commonly referred to as 'nanorods'. We believe that the term 'nanorods' is appropriate in this context, given that one of the dimensions (diameter) can indeed be below 100 nm.
- Again, in lines 302 to 304 it is not clear whether you are using alexa-tagged antibodies or alexa dye only.
Reply: We have revised the text in lines 302 to 304 to a goat anti-mouse IgG tagged with Alexa Fluor® 488.
- From the graph in Figure 4, it can be observed that the improvement in the fluorescent signal between GPTMS-modified glass and GPTMS-modified ZnO NRs is approximately 25%. Have the authors considered evaluating the use of GPTMS-modified glass without ZnO NRs? Perhaps the limit of detection would not be significantly worse, and the approach would benefit in terms of simplicity, cost-effectiveness, and feasibility.
Reply: Your suggestion to consider using GPTMS-modified glass without ZnO NRs is interesting. We acknowledge that a 25% improvement in the fluorescent signal may not seem substantial at first glance. However, the improvement is largely attributed to the large surface area provided by the ZnO NRs, which is a significant advantage in biosensing applications. This improvement could be crucial in scenarios where high sensitivity is required, such as low-abundance target detection.
While we agree that a more straightforward, more cost-effective approach has its merits, the added sensitivity provided by ZnO NRs could be a decisive factor in specific applications. Nonetheless, your comment has inspired us to consider further studies comparing the two approaches in terms of sensitivity, cost-effectiveness, and feasibility.
- From my perspective, the authors should articulate more clearly the significance of employing microfluidics in the procedure. Some of the steps appear to be conducted at stopped flow, suggesting that the procedure might also be executed without microfluidics. Could the authors please specify the advantages of utilizing microfluidics over a conventional batch procedure?
Reply: We appreciate your suggestion to elaborate on the significance of employing microfluidics in our procedure. You are correct that some steps are conducted at stopped flow; however, the use of microfluidics offers several advantages over conventional batch procedures. Specifically, microfluidic platforms allow for a reduction in sample volume, faster analysis time, and enhanced portability. These benefits are particularly relevant for the detection of dengue virus, where rapid and efficient diagnosis is crucial.
- Sentences in lines 358 – 360 are confusing: “The limit of detection (LOD) for the DENV-3 immunofluorescence assay, implemented on the ZnO NRs within the microfluidic platform, was ascertained to be 3.1 × 10⁻⁴ ng/mL, which is five-fold higher than the intensity recorded for the negative control.” Are the authors suggesting that the LoD is determined as the concentration corresponding to a fluorescence intensity that is five times that of the negative control? This approach does not align with the procedure recommended by IUPAC. Please refer to the 'limit of detection' definition in the IUPAC Gold Book. (https://goldbook.iupac.org/terms/view/L03540)
Reply: We establish a Threshold for LOD by our data in Figure 6B and calculate the LOD threshold as:
LOD threshold = Mean of control + (3 x standard deviation of control)
LOD threshold = 4.75 x 107 + (3 x 2.88 x 106)
LOD threshold = 5.6 x 107
Identify the Antigen Concentration for LOD:
The LOD is the lowest concentration at which the measured intensity (minus its standard deviation) is greater than the LOD threshold.
According to the data:
For 3.10 x10-4 ng/mL, the intensity minus SD is 9.45 x107 - 4.46 x106 = 9.00 x107, which is above threshold of 6.05 x107. Based on the presented data, the limit of detection (LOD) is at or less than 0.00031 ng/ml.
- Table 1 is very informative, although the variety in analytes and units used for the limit of detection makes a proper comparison challenging. Nonetheless, I would suggest that the authors express their LoD in PFU (instead of FFU) and in ng/mL, as these units are more commonly represented in the table.
Reply: We appreciate your suggestion for standardizing the units used in Table 1. We have revised the table to express the limit of detection (LoD) in PFU and ng/mL, as these units are more commonly used and will facilitate easier comparison.
- There are comments in the manuscript in page 4 by Sp1 and Sp2 which must be removed.
Reply: We have addressed and removed the comments and ensured that the manuscript is polished for publication.
- Some of the references seem to be mixed-up. For example, reference 1 is cited for “Notable examples include the Ebola virus, SARS, MERS, Zika, and most recently, SARS-CoV-2, the causative agent of COVID-19”. However, reference 1 deals with Dengue and does not mention SARS, MERS or Zika; similarly reference 13 deals with trophoblast detection with chips but is not too correlated to virus detection with microfluidics. Please, check the references again.
Reply: We have revised the references to better align with the context in which they are cited. Specifically, for the mention of Ebola, SARS, MERS, and Zika, we have added the following references:
- Molecular Evolution of Zika Virus during Its Emergence in the 20th Century. (DOI: 10.1371/journal.pntd.0002636)
- SARS: clinical features and diagnosis (DOI: 10.1046/j.1440-1843.2003.00520.x)
- SARS and MERS: recent insights into emerging coronaviruses (DOI: https://doi.org/10.1038/nrmicro.2016.81)
For the context where microfluidic platforms are discussed in relation to virus detection, we have replaced reference 13 with:
- Rapid and Fully Microfluidic Ebola Virus Detection with CRISPR-Cas13a (https://doi.org/10.1021/acssensors.9b00239)
- Portable microfluidic impedance biosensor for SARS-CoV-2 detection (https://doi.org/10.1016/j.bios.2023.115362)
We believe that these revisions address the concerns raised by the reviewers and enhance the scientific rigor and clarity of our manuscript. We are confident that the revised manuscript now meets the journal's standards and look forward to its successful publication.
Thank you for your time and consideration. We remain open to any further suggestions or revisions that may be necessary.
Best regards,
Sakon Rahong

Reviewer 3 Report
This paper reports ZnO nanorods to achieve Dengue virus detection. The designed materials are discussed, and detection performance is studied. This paper provided sufficient data on the material part, but more discussion and tests of the detection part should be provided. Therefore, I suggest publishing in Nanomaterials after major revision.
1. In the Introduction part, the authors claimed that ZnO nanorods have a high surface-to-volume ratio, but no such evidence applied, I recommend doing the BET test to prove this statement.
2. The equation for calculating LOD should be given, how do the authors determine the LOD of the detection? What’s more, the detection linear range should also be provided, which is an essential detection factor.
3. More optimization factors of detection assay should be given other than the substrate, like the operation time, wash times of PBTS, the amount of loading antibodies, etc.
4. Detection in real samples should also be studied to investigate the assay’s practical application and the specificity of the microfluidic platform should also be discussed.
5. The original comments (I believe between the authors) that come with the file should be deleted clean.
English still needs to be polished. And some typos should be corrected.
Author Response
Dear Reviewer,
We would like to express our sincere gratitude for the opportunity to revise our manuscript entitled " Metal Oxide Nanostructures Enhanced Microfluidic Platform for “Efficient and Sensitive Immunofluorescence Detection of Dengue Virus” Manuscript ID: nanomaterials-2646360. We are thankful to the reviewers for their constructive feedback, which has been instrumental in enhancing the quality and clarity of our work.
Below, we provide a point-by-point response to the referees' comments and detail the revisions made to the manuscript:
Reviewer-3
This paper reports ZnO nanorods to achieve Dengue virus detection. The designed materials are discussed, and detection performance is studied. This paper provided sufficient data on the material part, but more discussion and tests of the detection part should be provided. Therefore, I suggest publishing in Nanomaterials after major revision.
- In the Introduction part, the authors claimed that ZnO nanorods have a high surface-to-volume ratio, but no such evidence applied, I recommend doing the BET test to prove this statement.
Reply: We appreciate your suggestion to use the BET test to substantiate our claim regarding the high surface-to-volume ratio of ZnO nanorods. However, conducting a BET test currently is difficult for us. Instead, we have referred to the previous reports (reference number 29, 30 and 39) and performed surface-to-volume ratio calculations to confirm our claim. We believe that this approach provides credible evidence to support our statement.
Calculation
For a ZnO nanorod with a hexagonal structure, let's assume that the distance from one corner to the opposite corner (the diameter) is 100 nm with 5 mm length. The side length a can be calculated as
Surface Area of Hexagonal Face Ahex:
Lateral Surface Area Alateral:
Total Surface Area A :
Volume V :
Surface-to-Volume Ratio S/V:
Since the density of ZnO NRs is 1.3 ´ 109 /cm2, the surface area of the ZnO NRs in 1 centimeter square:
The surface area of a ZnO nanorod is much higher than that of a 1 cm cube, which has a surface area of 6 cm².
- The equation for calculating LOD should be given, how do the authors determine the LOD of the detection? What’s more, the detection linear range should also be provided, which is an essential detection factor.
Reply: We establish a Threshold for LOD by our data in Figure 6B and calculate the LOD threshold as:
LOD threshold = Mean of control + (3 x standard deviation of control)
LOD threshold = 4.75 x 107 + (3 x 2.88 x 106)
LOD threshold = 5.6 x 107
Identify the Antigen Concentration for LOD:
The LOD is the lowest concentration at which the measured intensity (minus its standard deviation) is greater than the LOD threshold.
According to the data:
For 3.10 x10-4 ng/mL, the intensity minus SD is 9.45 x107 - 4.46 x106 = 9.00 x107, which is above threshold of 6.05 x 107. Based on the presented data, the limit of detection (LOD) is at or less than 0.00031 ng/ml.
- More optimization factors of detection assay should be given other than the substrate, like the operation time, wash times of PBTS, the amount of loading antibodies, etc.
Reply: We have revised this section to provide a more comprehensive and clear description of the experimental procedures (line 209-226). Specifically, we have clarified the duration of 54 mAb flow, whether the flow was stopped during incubation, and how the system was washed between the two incubation steps. We have also provided details on the introduction of the DENV-3 sample, including the flow duration, volume injected, and whether incubation occurs while the sample is flowing.
“Conjugation with 1 µg/mL solution of 54 monoclonal antibody (54 mAb) was performed by introducing at a flow rate of 120 µL/min for 2 minutes. Subsequently, flow was halted, and the sample was incubated at room temperature for 30 minutes. The system was then washed with a 0.1% Tween-20 in phosphate-buffered saline (PBST) (Sigma-Aldrich, Inc.) at a flow rate of 120 µL/min for 2 minutes. This was followed by a nonspecific binding step using a 5% (w/w) bovine serum albumin (BSA) solution, also at a flow rate of 120 µl/min for 2 minutes. Another incubation period of 30 minutes was implemented before washing the system again with 0.1% PBST for 2 minutes. The Dengue Virus serotype 3 (DENV-3) was then introduced at varying concentrations, ranging from 3.1×10-4 to 3.1×101 ng/ml, with a consistent flow rate of 120 µl/min for 2 minutes. This was followed by a 30-minute incubation period and a subsequent wash with 0.1% PBST for 2 minutes. The 4G2 monoclonal antibody was then flowed at a concentration of 1 µg/ml and a flow rate of 120 µl/min for 2 minutes, followed by a 30-minute incubation. The system was then washed with 0.1% PBST for 2 minutes. Finally, the goat anti-mouse IgG tagged with Alexa Fluor® 488 (Prima Scientific Co., Ltd.) was introduced at a flow rate of 120 µl/min for 2 minutes, followed by a 15-minute incubation period and a final wash with 0.1% PBST for 2 minutes to remove any unbound secondary antibodies. The chip was then ready for fluorescence imaging.”
- Detection in real samples should also be studied to investigate the assay’s practical application and the specificity of the microfluidic platform should also be discussed.
Reply: We agree that these aspects are crucial for demonstrating the practical applicability of our assay. While we have not included these elements in the current manuscript, we have plans to conduct tests with real samples and will report these findings in future work. For the specificity of the microfluidic platform, we would like to clarify that the herringbone structure of our microfluidic platform enhances the likelihood of antigen-antibody binding on the ZnO NRs surface (line 358-361). This feature allows for rapid detection in a minimal sample volume, thereby increasing the specificity of our assay.
- The original comments (I believe between the authors) that come with the file should be deleted clean.
Reply: We have addressed and removed the comments and ensured that the manuscript is polished for publication.
We believe that these revisions address the concerns raised by the reviewers and enhance the scientific rigor and clarity of our manuscript. We are confident that the revised manuscript now meets the journal's standards and look forward to its successful publication.
Thank you for your time and consideration. We remain open to any further suggestions or revisions that may be necessary.
Best regards,
Sakon Rahong

Round 2
Reviewer 2 Report
Dear authors:
I am happy to know that some of my suggestions have helped you to improve your manuscript. From my point of view it is now clearer. Nonetheless, I would recommend you to check if text in line 218 is right (3.1×10^-4 to 3.1×10^1 ng/ml: Why not 3.2·10^-4 to 31 ng/mL? Is may be a minus sign missing?). Similarly, text in line 243 (the wave number range of 4,000-4000 cm-1) sounds strange. Isn't 4,000 and 4000 the same number? May be there is an extra zero in the second one?
Reviewer 3 Report
The author made good revision, no comment anymore.